# Role of Somatostatin in the Regulation of Central and Peripheral Factors of Satiety and Obesity

**DOI:** 10.3390/ijms21072568

**Published:** 2020-04-07

**Authors:** Ujendra Kumar, Sneha Singh

**Affiliations:** Faculty of Pharmaceutical Sciences, The University of British Columbia, Vancouver, BC V6T 1Z3, Canada; sneha.singh@alumni.ubc.ca

**Keywords:** appetite, obesity, satiety, somatostatin, somatostatin receptors

## Abstract

Obesity is one of the major social and health problems globally and often associated with various other pathological conditions. In addition to unregulated eating behaviour, circulating peptide-mediated hormonal secretion and signaling pathways play a critical role in food intake induced obesity. Amongst the many peptides involved in the regulation of food-seeking behaviour, somatostatin (SST) is the one which plays a determinant role in the complex process of appetite. SST is involved in the regulation of release and secretion of other peptides, neuronal integrity, and hormonal regulation. Based on past and recent studies, SST might serve as a bridge between central and peripheral tissues with a significant impact on obesity-associated with food intake behaviour and energy expenditure. Here, we present a comprehensive review describing the role of SST in the modulation of multiple central and peripheral signaling molecules. In addition, we highlight recent progress and contribution of SST and its receptors in food-seeking behaviour, obesity (orexigenic), and satiety (anorexigenic) associated pathways and mechanism.

## 1. Introduction

Somatostatin (SST), also known as somatotropin release-inhibiting factor, is a growth hormone inhibitory peptide that was first discovered in the hypothalamus in 1973 [1]. SST distribution is not restricted to the hypothalamus; instead, it is widely distributed in the central nervous system (CNS) and many other peripheral tissues including pituitary, pancreas, thyroid, and importantly in the gastrointestinal tract (GIT) in a tissue and site-specific manner [2,3]. SST acts as a classical endocrine hormone, a local regulator, as well as a true neurotransmitter and neuromodulator [4]. Besides its multiple endocrine and CNS actions, almost all GIT functions are regulated by SST. In the GIT, SST is released by luminal acid stimulation and regulates the release of gastric acid by a negative feedback mechanism [5]. Compared to carbohydrates, fat and protein are the major stimulants that release SST [6]. SST with its widespread distribution acts in a distinct manner in target tissues via binding to five different receptor subtypes: SST receptors 1–5 (SSTR1–5) [3]. SSTR subtypes belong to the family of G-protein coupled receptors (GPCRs), with seven transmembrane proteins. SSTRs coupled to inhibitory G protein (Gi) and inhibit adenylyl cyclase and suppressed intracellular cyclic adenosine monophosphate (cAMP) [2,3,7]. The use of natural SST in medicine is limited owing to its short half-life (1–3 min) after release and rapid degradation by peptidases in plasma and tissues [8]. Therefore, many novel and stable long-acting analogues have been developed to enhance the efficacy and receptors selectivity and specificity. Synthetic analogues are structurally similar to natural SST and are currently used in the treatment of several pathological conditions including acromegaly and neuroendocrine tumours and have potential clinical applications in the management of inflammation and nociception, obesity, and diabetic complications [9,10,11,12,13]. In addition to SST analogues, SST vaccinations, namely JH17 and JH18, using intraperitoneal route of administration have been reported in reduction of weight gain and body weight percentage of normal, non-obese mice and mice with diet-induced obesity [14].

SST mediated regulation of obesity and satiety is most controversial and varies in a species- and dose-specific manner. Impaired energy regulation, in general, is believed to be the leading cause of obesity. The concept that SST might involve in obesity emerged from the regulation of insulin release [15,16]. Hyperinsulinemia is often seen in obese individuals and postprandial insulin resistance is a critical determinant of obesity in children [17]. Insulin that promotes energy storing in adipose tissue is considered as the prime suspect of obesity and the target of therapy [18,19]. In the treatment of hypothalamic obesity, SST and its analogues limit the release of insulin and inhibit adipogenesis [20]. SST analogue octreotide (OCT) alters human gastric functions such as stomach emptying and stomach volume and suppresses insulin secretion, which is beneficial in the management of obesity [21]. Therefore, pharmacological approaches to alter satiation and insulin secretion may have an impact on metabolic disorders such as obesity [22,23]. 

SST and its analogues block the insulin secretion from pancreatic β-cell and seem, by all accounts, to be a promising agent to treat obesity [9,24]. However, insulin is not the only hormone that is regulated by SST; other orexigenic and anorexigenic peptides are also regulated by SST centrally or peripherally with a significant effect on appetite and energy regulation. The underlying molecular mechanisms of SST mediated anti-obesity role and its five receptor subtypes are not well understood. Accordingly, understanding the role of SST and the neurohormonal pathways involved in satiety and obesity may assist with developing novel SST analogues and potential treatment strategies [13,25,26]. The role of SST in food intake and appetite has been reviewed previously; however, SST mediated regulation of other critical peripheral determinants of food-seeking behaviour is not well understood. In the present review, we explore the central and peripheral mechanisms and roles of SST in the regulation of satiety and appetite. 

## 2. Obesity and Satiety

Obesity is an imminent global pervasive with the increasing prevalence and repercussions such as multiple aetiologies, which subsequently leads to pernicious effects on normal physiological functioning [27]. In the last decade, obesity, which is associated with excess food intake, has become a global crisis and major concern of healthcare systems [28,29,30]. At present, 500 million people worldwide are living with obesity [31,32]. By 2030, it is projected that there will be 1.35 billion and 573 million overweight and obese adults, respectively [33]. Studies propose that obesity is a complex co-morbid health concern and one of the leading causes of metabolic and endocrine disorders [34,35,36,37,38,39,40,41,42,43,44,45,46,47]. Not only genetic susceptibility but also environmental factors are responsible for obesity, which may account for the reduction in metabolic rate, sedentary lifestyle, and reduced activity of the sympathetic nervous system [48]. All these interconnected events collectively cause an imbalance in the regulation of energy homeostasis, which eventually leads to obesity. This imbalance usually includes the inability to lower intake of calories along with having an ample amount of physical exercise/activity to curtail the effect of the ingested calories. Thus, caloric intake and energy expenditure must be considered to regulate homeostasis [49]. A tenuous imbalance of less than 0.5% in caloric intake over expenditure is enough to cause an increase in weight [50]. 

At present, the pharmacological interventions available for the regulation of food intake and obesity are inadequate and unable to collectively address the clinical diversity of various side effects and contraindications. The number of drugs that have been approved for weight reduction are very few [51]. It is interesting to note that recent observations on circulatory peptides, steroids, and associated signaling molecules, which primarily elicit their effect on the hypothalamus, brain stem, and autonomous system, have changed our perception of food intake [52]. Furthermore, these hormones are produced and secreted by fat cells, GIT, and the pancreas. Amongst them, SST plays a prominent role in food-seeking behaviour.

Contrary to obesity, satiety is a sense of fullness due to ephemeral lack of interest in further ingestion of food, which eventually leads to the suppression of appetite. Fullness is supposed to have a vital role in weight management because it will impact the frequency and amount of food and drink consumed later [53]. The feeling of satiety is a well-organized and biologically oriented process [54,55]. Satiety is a well-articulated process which reduces gastric distension and cholecystokinin (CCK) release in gut post ingestion of nutrients and importantly has been associated with signals from periphery via vagal afferent fibers that terminate in nucleus of the tractus solitaries (NTS) in the brain stem [56]. Besides, leptin, a hormone secreted by adipocytes, is also associated with satiety signals and studies further indicate that satiety response to CCK that was blocked in leptin receptor-deficient rat was regained following restoration of the leptin receptor in arcuate nucleus (ARC) [57,58,59]. Studies also support the role of NTS vagus afferent in response to leptin [60,61]. Feeling satiated will, in the long run, smother the inclination for cravings and lead to lesser consumption of food in the next meal. Satiety is not a response; it involves a complex mechanism for its operation, which includes the crucial role of the hypothalamus and peripheral hormones and determines energy expenditure [62]. Measurement of satiation can be done either directly by assessing food intake or indirectly by subjectively rating the urge for appetite [63]. Schwartz et al. demonstrated that hypothalamus is the central brain region responsible for adiposity signal; however, it is not an active participant in satiety [64]. It is interesting to note that most satiety associated information is relayed to NTS afferent fibers from the vagus nerve and afferent from GIT through the spinal cord. Mechanical or chemical stimulation, neural input, and hormonal release from peripheral organs such as liver, GIT, stomach, and adipose tissues during food intake initiate satiety signals. Moreover, it is believed that satiety signals mediated blockade of feeding can occur even if the communication between forebrain and hindbrain is interrupted [55,64,65]. Overall, it is comprehended that satiety plays a vital role in controlling food intake and body weight. 

### 2.1. Role of Hypothalamus in Obesity and Satiety 

The role of CNS, specifically the hypothalamus in food-seeking behaviour is well established [66]. Perturbed energy balance plays a determinant role in food-seeking behaviour and body fat is a critical player to make the final decision whether it is food intake or energy expenditure. In this context, content of circulating signals (specifically peptides and steroids emerged first from ob/ob and db/db mouse models) in response to fat stores was first proposed in 1953 by Gordon Kennedy and later investigations affirmed that hypothalamus is the fundamental region that plays a prominent role in regulation of anorexigenic and orexigenic behaviour [67]. Previous animal studies have supported that an intact hypothalamus is a prerequisite for the presence of satiety factor in obesity [52,67]. Hypothalamic brain regions including ARC, ventromedial nucleus (VMN), periventricular nucleus (PeVN), paraventricular nuclei (PVN), and lateral hypothalamic area (LHA) are well connected and associated with satiety and food-seeking behaviour. 

The distribution of neuronal cells in the different hypothalamic regions with selective and preferential expression of hormonal peptides with orexigenic and anorexigenic stimuli attested the hypothalamus as a key centre in the regulation of appetite and energy homeostasis [68]. Previous studies with disrupted different hypothalamic regions implicate the importance and significance of the hypothalamus in food intake and satiety [69]. In the hypothalamus, ARC is composed of orexigenic neurons producing neuropeptide Y (NPY) and agouti-related peptide (AgRP) and anorexigenic neurons, which synthesize proopiomelanocortin (POMC) and cocaine- and amphetamine-regulated transcript (CART). ARC is not fully protected by the blood–brain barrier (BBB) and is believed to integrate most peripheral signals of leptin, insulin, and ghrelin as well as nutrients via systemic circulation and implicated in food-seeking behaviour. Orexigenic and anorexigenic neurons confined in ARC also project to other hypothalamic regions including VMN, PVN, and LHA, as well as to the brain stem and spinal cord [70,71]. The key regulators of catabolic response, including corticotropin-releasing hormone (CRH), thyrotropin-releasing hormone, vasopressin, and orexin (OX), are synthesized and released from neuronal cells present in PVN and exert an inhibitory role on food intake and weight gain [72]. Animals with disrupted VMN are prone to eat excess food that causes hyperphagia, hyperglycemia, and obesity, proposing that VMN is primarily involved in satiety and glucose homeostasis. VMN receives neuronal input from ARC and sends projection to ARC. In particular, VMN additionally contains neurons that produce brain-derived neurotrophic factor (BDNF) that is associated with anorexigenic behaviour [73]. Several interconnected pathways between LHA and other CNS regions suggest that LHA serves as a key centre in the regulation of feeding behaviour [66,74,75]. Furthermore, the lesion of LHA prompts anorexia and weight loss [65,76]. Dorsomedial nucleus (DMN) received NPY and melanocyte stimulating hormone (MSH) input from ARC and damage to DMN are associated with hyperphagia and obesity. LHA also contains melanin-concentrating hormone and OX producing neurons and disruption resulted in hypophagia and weight loss. DMN and LHA are believed to have neuronal cells which sense leptin and glucose [77,78]. Recent studies have added new information regarding the expression of several other neurotransmitters and peptides and their coexpression with NPY/AgRP and POMC in different regions of hypothalamus. 

Most neuroendocrine signals and functions displayed by central and peripheral stimuli in regulation and maintenance of appetite are confined in different regions of the hypothalamus. Amongst them ARC is the hypothalamic nuclei that maintains the balance between food intake and energy requirement [15]. ARC remains vulnerable to endocrine signals coming from the rest of the body due to the fenestrated capillaries and close proximity to median eminence [79]. ARC is composed of different types of neurons that respond to endocrine signals and produce impulses for the regulation of obesity and satiety. In the hypothalamus, two major types of neurons including anorexigenic and orexigenic are confined in different nuclei, including ARC, PVN, and LHA. Previous studies have shown that anorexigenic neurons of ARC coexpressed two important peptides, which reduce hunger and produce satiety [16,69,80,81]. The first peptide is POMC, a precursor peptide which is cleaved by propeptide convertases 1 and 2 at lysine and arginine residues. Furthermore, POMC is involved in the formation of many active neuropeptides like α-MSH, which binds to melanocortin 3 (MC3R) or melanocortin 4 receptor (MC4R) and leads to satiety [82]. Thus, the lack of POMC binding at MC4R or mutations in propeptide convertases 1 can lead to early-onset of obesity [16,69,79,80,81]. The second peptide is CART that produces anorexic effects [83]. Furthermore, two other peptides, namely NPY and AgRP coexpressed in orexigenic neurons of ARC, and associated with increased appetite. NPY binds to different neuropeptide Y receptor 1–6 (NPYR1–6) member of GPCR family [80]. However, the binding of NPY to NPYR1 and NPYR5 act as an antagonist on MC3R and MC4R receptors, respectively, thereby increasing appetite and leading to obesity [84], as described in Figure 1. Similar to NPY, AgRP also stimulates food intake [16,69].

In addition to POMC, the presence of leptin receptors in the hypothalamus contributes significantly to the regulation of food intake and energy balance. The perturbed leptin signaling is often seen in obese person and ob/ob and db/db mouse model devoid of leptin and leptin receptor gene [85,86]. It is interesting to note that leptin is not restricted to the hypothalamus, but also present in the hippocampus, which principally is involved in learning and memory, and similarly contains leptin receptor and associated with energy regulation [87,88]. The authors further demonstrated increased body weight and food intake in rats with lesion of hippocampus. Furthermore, developing diet-induced obesity often resulted from the failure of leptin mediated signaling in POMC and AgRP neurons in ARC. Previous studies have shown that the absence or inhibition of AgRP neurons in ARC leads to suppressed the food intake, whereas activation prompts food-seeking behaviour even in well-fed mice [89,90,91,92,93]. Recently, the presence of inhibitory GABAergic and excitatory glutamatergic neurons in LHA with opposing effects on food-seeking and satiety added new neuronal circuits in the regulation of food intake [66,94]. Furthermore, GABA released upon activation of AgRP/NPY neurons inhibits POMC neurons in hypothalamus and resulted in suppressed food intake [95]. Recent studies have included a new brain region that might play a crucial role in the regulation of food intake and energy balance. In this context, Schneeberger et al. described the role of GABAergic neurons in the dorsal raphe nucleus linked to the regulation of energy expenditure via interfering in locomotion and thermogenesis [96]. This information adds a new dimension for the role of dorsal raphe nucleus associated neuronal circuit in the regulation of energy expenditure. Schneeberger et al. also found the opposing role of activation (decreased energy expenditure) associated with thermogenesis and locomotor activity, whereas inhibition (increased energy expenditure) was linked to only locomotor activity without any involvement of thermogenesis [96]. Serotonin, a prominent neurotransmitter in CNS, is also a major regulator of appetite and promotion of energy expenditure in addition to its association in behaviour and anxiety [97,98]. In addition, peripheral serotonin prevents obesity and insulin resistance [95]. The changes in circadian rhythm and meal induced release of circulating serotonin have been implicated with increased release of serotonin into circulation from GIT in condition such as obesity and type 2 diabetes [95,99]. Conclusive evidence in support of appetite regulation emerged from the studies showing the inhibition of AgRP/NPY and stimulation of POMC neurons by hypothalamic serotonin in a similar manner as seen with the use of leptin [98,100,101]. Moreover, serotonin receptor agonists are potential therapeutic alternatives in treatment of obesity.

The SST positive neurons in the hypothalamus are mostly confined to the PeVN and account for the major portion of SST immunoreactivity in hypothalamus [102]. In addition, SST positive neurons are also present in PVN, suprachiasmatic nucleus, ARC, and VMN [103,104]. Importantly, SST presence in hypothalamic nuclei and the regulation of growth hormone-releasing hormone (GHRH), CRH, and inhibition of ghrelin release attest its role in appetite [3,105,106]. SST might inhibit POMC because SSTR subtypes are also coexpressed in neurons expressing POMC mRNA in ARC. In addition to prominent hormones, endocannabinoids are implicated in appetite and serve as orexigenic stimuli via activation of cannabinoids receptor 1 (CB1R) in the brain specifically in the hypothalamus. Although CB1R expression in the hypothalamus is relatively less, we recently described CB1R positive neurons in the different nucleus of hypothalamus, including PVN, PeVN, VMN, and ARC, as well as innervated nerve fibers in the median eminence [104]. These results are consistent with previous studies and support the role of CB1R in regulating food and energy balance, probably through the hypothalamus [107,108,109]. In the CNS, SST is also associated with neuronal nitric oxide synthase (nNOS). In the hypothalamus, nNOS plays a crucial role in food intake and energy balance in addition to several physiological functions. The association of nNOS in the regulation of appetite is through the modulation of well-appreciated modulators of food intake including NPY, ghrelin, and leptin. In support of these observations, we recently demonstrated the subcellular distribution of nNOS in the hypothalamus including PVN, PeVN, VMN, and ARC [104].

### 2.2. Role of Peripheral Hormones on Obesity and Satiety

Food-seeking behaviour does not always depend on CNS stimuli; the peripheral system including GI and adipose tissue also plays a prominent role. Endogenous gut hormones such as CCK, glucagon-like peptide 1 (GLP-1), oxyntomodulin, peptide tyrosine, and pancreatic polypeptide have anorexigenic effect in humans which decrease food intake. Conversely, ghrelin, an appetite modulator, has an orexigenic effect in humans and leading to increased food intake (Figure 1). On the other hand, an increased peripheral insulin level is associated with obesity through the adipostat mechanism [20,110].

Physiochemical properties of the ingested food stimulate the secretion of various hormones from the lining of the GIT. GI hormones are also expressed in the CNS, which relay metabolic information between the GIT and the brain, influencing meal initiation and termination. The majority of the gut hormones that act on various sites such as the hypothalamus, brainstem, and vagus afferent decrease the food intake (Table 1). Ghrelin is an amino acid peptide hormone produced from the fundic region of the stomach and has been identified as an orexigenic gut hormone [65,111,112,113]. In peripheral tissues, the resistance to insulin linked with obesity is perhaps the reason for the development of insulin resistance in the hypothalamus. It is yet to be determined whether neuronal resistance to insulin is associated with obesity.

### 2.3. Link between the Hypothalamus and Peripheral Hormones in the Regulation of Obesity and Satiety

In contrast to the well-accepted role of the hypothalamus, food intake is also regulated by the peripheral hormonal system either through the vagus nerve or by crossing the BBB [128]. The dorsal vagal complex (DVC) of the brainstem consists of three different regions including the area postrema, NTS, and the dorsal motor nucleus of the vagus nerve [129]. Further, it is conceivable that peripheral signals of food intake are linked to hypothalamic nuclei through the DVC. Sensory information from the gut is directly carried by afferent vagal nerves to the NTS. In addition, neural projections exist between brainstem and hypothalamus [79]. The mesolimbic reward system including the ventral tegmental area (VTA) and the nucleus accumbens (NAc) are also connected to the hypothalamus and regulate the hedonic behaviour of food intake [69,129]. During fasting state, when the glucose levels drop, the activation of glutamatergic and orexinergic neurons in LHA subsequently activates dopamine (DA) neurons in VTA. This eventually results in activation of inhibitory GABAergic neurons in LHA projecting to VTA, thus leading to increased food intake (caloric reward) [130,131,132].

Leptin, a satiety hormone secreted from adipocytes, is linked to appetite, energy homeostasis, and glucose metabolism and stimulates POMC and CART expressing neurons directly in the hypothalamus [133]. Body fat is well associated with leptin levels and an excess level of leptin due to hyperleptinemia is a characteristic of obesity [134]. In addition to stimulating POMC expressing neurons, leptin also inhibits AgRP positive neurons and inhibits food intake. Fasting state reduces leptin expression in adipose tissue and its plasma concentrations, which cause the upregulation of the NPY mediated increase in food intake [80]. Similar to leptin, AgRP and NPY expression is reduced and POMC synthesis is increased by insulin with suppressed food intake [15]. Another peripheral hormone, ghrelin, increases food intake and deposition of fat by increasing NPY and AgRP and reducing POMC expression [135]. It suggests that CNS is linked to the gut through the endocrine pathway and maintains appetite in association with peripheral hormones (Figure 1). Similar to leptin, other satiety associated gut peptide hormones such as GLP-1, oxyntomodulin, and peptide YY also play a critical role in food-seeking behaviour (Table 1).

## 3. Distribution of Somatostatin and Somatostatin Receptors

SST is widely distributed in most parts of the body. The CNS, GIT, and endocrine glands are the major sites of SST production and action [2]. SST is derived from a common precursor, preprosomatostatin. Initially, preprosomatostatin is converted to prosomatostatin through tissue-specific posttranslational processing. Prosomatostatin is further processed into two biologically active isoforms: somatostatin-14 (SST-14) and somatostatin-28 (SST-28). A disulphide bridge is present in both forms of SST, which gives a cyclic structure [136,137]. SST-14 is the isoform that was initially isolated and identified as an inhibitor of growth hormone release. A few years later, SST-28 was isolated from the porcine gut as an extended SST-14 sequence to the amino terminus [138,139,140]. SST-28 is the predominant form in the retina and intestinal mucosal cells, whereas SST-14 is the prominent form in the CNS and in most peripheral organs, including the GIT [141]. In addition to the expression in the endocrine tissues such as pancreas and pituitary, relatively less expression of SST is also seen in thyroid, adrenals, kidney, prostate, and blood vessels, as well as in immune cells [2]. SST is also expressed in mucosal D cells of the GIT, which are dispersed in different regions of GIT including the stomach, intestine, and enteric nervous system [66,142]. In rats, GIT contains the most amount of SST and plays multiple roles. In GIT, all the hormones with significant diversity in physiological functions are inhibited by SST. SST’s role in motor activity of GIT further results in inhibition of gastric emptying, contraction of the gall bladder, and segmentation of the intestine [2]. The prominent role of SST proposed is the inhibition of GH directly and via inhibition of GHRH indirectly [143,144]. Previous studies have shown a profound stimulatory role of DA, glucagon, CCK, acetylcholine, alpha-adrenergic agonists, vasoactive intestinal peptide, and substance P, whereas glucose exerts inhibitory role on the secretion of SST from the hypothalamus [2,145,146,147,148]. Besides its role as a true neurotransmitter and neuromodulator in the CNS, as well as the inhibition of hormones from several endocrine tumours, SST also leads to the shrinkage of these tumours by inhibition of cell proliferation through apoptosis [149]. 

### 3.1. Somatostatin Receptors

Soon after the discovery of SST-14 and SST-28 and widespread distribution in different body parts, five different receptor subtypes, namely SST receptors 1–5, were identified, which are encoded by five different genes and exhibit 70% structural similarity between human and rodents. SSTR subtypes belong to a family of GPCRs and all five SSTR subtypes have been pharmacologically characterized [150]. In addition, to display the classical features of GPCRs, SSTRs also contain postsynaptic density protein, disc large, and zona occludens-1 (PDZ) domain, and many other interacting proteins have also been identified in a receptor-specific manner [2,151]. Both SST-14 and SST-28 have shown a high affinity towards human SST receptor subtypes 1–4. However, the binding affinity of SST-28 towards SSTR5 was found to be 5–10 times more than for SST-14 [2,150]. Several synthetic analogues have been developed with improved half-life and better efficiency and effectiveness. Such long-acting analogues were primarily formulated for the treatment of tumours of different origin including GIT, pancreas, gut, and pituitary [152,153]. All five SSTR subtypes are widely distributed throughout the CNS in a receptor and brain region-specific manner. In the pancreas and pituitary, SSTR1–5 are well expressed and exhibited receptor-specific coexpression with hormone-producing cells [154]. In addition, SSTR subtypes are also expressed in peripheral tissues. The expression of SSTRs in cancer tissues has drawn significant attention towards cancer treatment as a potential therapeutic target. SSTR subtypes have also been detected at the level of mRNA and protein in GIT in a species and receptor-specific manner. SSTR2 and SSTR5 are the predominant subtypes in endocrine tissues [155,156,157]. 

The divergent roles of SST in the target tissues are either direct or indirect. Direct biological actions of SST are mediated through five different receptor subtypes, namely SST receptors 1–5 (SSTR1–5). SSTR subtypes couple to inhibitory G-protein and exert inhibitory role on adenylyl cyclase and are consequently involved in inhibition of the second messenger cAMP. SSTR subtypes play a crucial role in receptor and tissue-specific manner [2]. SSTR1 triggers anti-secretory effects on growth hormone, prolactin, and calcitonin. SSTR2 also mediates inhibitory action on the secretion of growth hormone, glucagon, insulin, interferon-γ, and gastric acid [158,159]. SSTR3 reduces cell proliferation and induces cell apoptosis [160]. SSTR4 is associated with pain and inflammation [11,13,161]. A recent study has also demonstrated the role of SSTR4 in anxiety and depression [162]. SSTR5 has an inhibitory effect on growth hormone, adrenocorticotropic hormone (ACTH), insulin, and GLP-1, and it inhibits the secretion of amylase. A summary of the physiological actions of SST and associated receptor subtypes in target sites is shown in Figure 2. In addition to the anti-proliferative effect in peripheral tissue, SST exerts a significant role in food intake and drinking behaviour via activation of specific receptors [26,121,122,163,164,165]. Besides the prominent role in the regulation of hormonal secretion, SSTR subtypes play a critical role in modulation of complex downstream signal transduction pathways [2,3]. It is worth mentioning here that the distribution of SSTR subtypes in the brain region such as hypothalamus involves appetite and energy homeostasis. It is now undisputed and well established that the hypothalamus is the brain region which is the major site of SST production and physiological actions. In rat hypothalamus, SSTR1–4 are well expressed, whereas SSTR5 is absent in all major regions of the hypothalamus except median eminence, which exhibited few cells positive to SSTR5 [166]. Conversely, all five SSTR subtypes are well expressed in mice hypothalamus [167].

### 3.2. Role of Somatostatin and Its Receptors in Satiety

The role of SST in the regulation of food intake and energy store and expenditure is most controversial. Previous studies support the pro- and anti-satiety effect of SST that depends on species, doses, and mode of administration [26,163,168]. It is not well understood if SST regulates satiety via direct or indirect mechanisms. The physiochemical properties of ingested food stimulate the secretion of various hormones from the GIT which are also present in the CNS and relay metabolic information between the GIT and the brain (Table 1). SST inhibits insulin and glucagon secretions from pancreas inhibits gastric emptying, and gallbladder contraction [169,170]. The immunohistochemical and in situ hybridization studies revealed that SSTR2 is the predominant receptor subtype in the human stomach with two splice variants, namely SSTR2A and SSTR2B [171]. SST induces satiety, which further results in decreased food intake [25]. The exact mechanism of SST associated with the regulation of satiety is not yet well understood; however, the proposed mechanisms for satiety are linked to delay in gastric emptying and gut motility, which exert direct intake including the hypothalamus, hippocampus, cortex and brain stem, is the first indication for the possible effects on appetite and adipocytes as well as in the modulation of gut hormones [172,173,174,175,176]. On the other hand, SST and its analogues may decrease satiety and increase food intake by inhibiting the release of endogenous satiety factors such as CCK and GLP-1 [25,63,177,178]. Stengel et al. reported that intracerebroventricular (icv) injection of stable SST agonists in rodents stimulates food intake and drinking behaviour centrally by the activation of SSTR2 [26]. 

### 3.3. Effect of Somatostatin and Its Analogues on Satiety

The presence of SST in the brain regions which are involved in the regulation of food intake displays the role of SST in appetite. SST and its analogues have been proven to induce satiety effect in children and adults (Table 2). SST prolongs intestinal transit time, and the SST analogue OCT induced a threefold prolongation of mouth to cecum transit time [179]. These motility interfering effects may result in satiety. In addition, SST and its analogues interfere with the absorption of meals, which may further contribute to the satiety effect [179,180,181,182]. Conversely, Lieverse et al. reported that, in the presence of a low dose of intraduodenal fat, SST may decrease satiety and increase food intake by inhibiting the release of endogenous satiety factors [25]. In contrast to the inhibitory actions of peripheral SST, centrally mediated actions of SST on satiety have also been reported in various animal studies. Stengel et al. reported that centrally administered SSTR2 antagonist reduced the cumulative dark phase food intake in rats [121]. The effects of SST and its analogues on satiety are shown in Table 2.

SST-14 and SST analogue OCT injection directly into the brain or by icv route in the hippocampus and cortex of the rat prompted food intake [184,187,188,189]. Moreover, these studies emphasized that SST effect is associated with doses and route of administration. In basal condition (light/dark phase), icv administration of low doses of pan-SST agonist, ODT8-SST or OCT, enhanced food intake in rats and/or mice in a very short time post-injection [163,168,185,188]. Such quick and rapid food-seeking nature is not limited to rat only, but it has also been observed in chicks [186]. In contrast to lower doses, icv administration of SST at relatively higher doses elicits anorexigenic effect in rats and mice [190,191,192,193,194]. The occurrence of dose associated opposing effects of SST was argued due to the impact of competitive behavioural changes or leakage of SST into peripheral circulation where SST is known to exert an inhibitory role [195,196,197]. Furthermore, enhanced food intake in response to icv injection of SST in rodents leads to activation of SST signaling in the brain. Additional findings reported that the administration of OCT promotes intake of food and leads to weight gain following bariatric surgery [198].

### 3.4. Somatostatin and Obesity 

The hypothalamus regulates glucose metabolism through parasympathetic activity. Hypothalamic damage, mainly due to brain tumour, may result in upregulation of parasympathetic activity resulting in hyperinsulinemia [199]. Often, obesity is associated with hyperinsulinemia, both under fasting conditions and post-prandially. Increased insulin secretion promotes adipogenesis by stimulating the activity of adipose tissue lipoprotein lipase resulting in weight gain and obesity [200]. SST and its analogues bind to SSTRs on the pancreatic β-cells and inhibit voltage-gated calcium-channels resulting in suppression of the early insulin response to glucose and thereby limiting the conversion of energy to adipose tissue [20,201,202,203]. The effect of SST analogues, OCT and lanreotide on food intake is also reported in hypothalamic obesity [20,204,205]. Efficacy of OCT has been proven beneficial in the treatment of obesity. SST analogues are approved for the treatment of acromegaly and carcinoid syndrome [206,207]. Although controversy exists, SST analogues have been used in diabetic retinopathy and hypothalamic obesity [208,209]. Currently, the efficacy of SST and its analogues in obesity is being evaluated in various clinical studies [9,20,210,211]. 

The hedonic eating behaviour of palatable food that is regulated by several cortical brain regions is a crucial contributor in obesity and satiety in a region-specific and neuronal phenotype manner. Whether SST plays any role in this direction is not well understood. Recently, Zhu et al. demonstrated that basal forebrain regions rich with GABAergic neurons play a key role in food intake [212]. This study also stated that the activation of the basal forebrain SST positive neurons selectively showed a preference for high-calorie food intake (high-fat chow and high sucrose water) within minutes with no effect on normal food. Furthermore, the projections from SST expressing neurons to the LHA induced high-fat intake without inducing preference for high sucrose. These observations also emphasized the increased fat and sucrose intake following optogenetic activation of SST neurons in basal forebrain and with anxiety-like behaviour in mice [212]. A recent study identified a subset of SST positive neuronal cells in ARC co-expressing AgRP, and the activation of these neurons was sufficient to drive feeding behaviour, although not selectively for fat, which suggests that different SSTR subtypes might be responsible for different feeding behaviours [213]. The safety and efficacy of SST and its analogues were evaluated in various clinical trials involving obese children and adults [13,120,209,214,215,216,217,218,219,220] (Figure 3).

### 3.5. Role of Somatostatin Receptor in Regulation of Food Intake Behaviour

Like SST five different SSTR subtypes also display widespread distribution in different parts of the brain, including the hypothalamus, and serve as a key central regulator of feeding behaviour in rodents and humans. The distributional pattern of SSTR subtypes is selective in a species-specific manner. We demonstrated significant variation in SSTR subtypes in hypothalamus. SSTR5 is absent in the hypothalamus of the rat but strongly expressed in the hypothalamus of mice [166,167]. Most importantly, increased mRNA expression of SSTR2 has also been described in a diet-induced rat model of obesity [221]. Furthermore, polymorphism in *SSTR2* gene has been associated with obesity and food intake in the Mediterranean population [222]. Thus, consistent with these studies, it is believed that SSTR2 is the prominent receptor subtype that plays a crucial role in food and water intake in the dark phase and, importantly, opposes the anorexic response to the visceral stressor. We postulate the role of SSTR5 in food-seeking approaches but in a species-specific manner.

## 4. Role of Somatostatin in Regulation of Brain-Derived Neurotrophic Factor Induced Appetite

SST expression and physiological functions in central and peripheral target are modulated by several factors, among them BDNF, a neurotrophic factor, affects its regulation [223,224]. However, the underlying molecular mechanisms describing the role of BDNF in SST mediated food-seeking behaviour and appetite are not well described. Here, we aim to describe functional relation between SST and BDNF in food-seeking behaviour. BDNF, a member of the neurotrophin family, is highly expressed in CNS and plays multiple roles, including development, synaptic neurotransmission, and plasticity via binding to high-affinity tyrosine kinase receptor B (TrkB). In the hypothalamus, BDNF is highly expressed in DMN, a centre of appetite. BDNF participates in the regulation of food intake and is a critical mediator of the anorexic effect of appetite regulators, including leptin, insulin, and pancreatic polypeptide [225,226,227]. The low level of circulatory BDNF is associated with a higher risk of eating disorder including anorexia nervosa and bulimia nervosa. Studies have shown low levels of BDNF in obese patients and those with diabetes type 2 [228]. Interestingly, the loss of BDNF in diabetic patients is independent of obesity, which indicates two different mechanisms in the regulation of obesity and insulin resistance by BDNF [229]. Two other molecular shreds of evidence with chromosomal inversion and child with perturbed TrkB receptor in hyperphagia supported by studies showing that BDNF haploinsufficiency is linked to hyperphagia and obesity [230,231]. The prominent role of BDNF in the regulation of food intake is further strengthened from observation by using icv infusion of BDNF that resulted in suppression of weight gain in rat and second from BDNF heterozygous mice displaying 50% loss of BDNF expression and age-dependent obesity [227,232,233,234]. BDNF-deficient mice are resistant to leptin and exhibit a high level of insulin.

The molecular mechanisms for the role of BDNF in food-seeking behaviour and its distribution in different region of the hypothalamus with high expression in VMN and interaction with orexigenic and anorexigenic are more complex than it seems, as reviewed by Rosas-Vargas et al. [225]. The exogenous infusion of BDNF reversed MC4R induced obesity and hyperphagia partially in agouti lethal yellow mice. Decreased BDNF expression in food-deprived mice is reversed by MC3/4R agonist, supporting the role of receptor in regulation of BDNF expression. Furthermore, an interesting observation from Komori et al. established a relation between BDNF and leptin and showed increased mRNA and protein expression of BDNF in VMN in response to iv administration of leptin [235]. These observations further emphasized that either leptin or leptin–receptor interaction and/or the presence of α-MSH in ARC activate BDNF via MC4R. Moreover, db/db mice treated with BDNF blocked hyperphagia and metabolic changes. Xu et al. demonstrated the role of MC4R and BDNF and its receptor TrkB in concert towards the regulation of energy balance [226]. Such a profound functional interaction between MC4R and BDNF was further supported by BDNF administration in mice lacking MC4R with consequent suppression of hyperphagia and weight [226]. These observations further explore the role of insulin in the stimulation of phosphoinositide 3-kinase in POMC expressing neurons in ARC or binding of insulin to its receptor in NPY/AgRP neurons in ARC. In the hypothalamus, another anorexigenic peptide which is regulated by BNDF is NPY. It is highly expressed and produced in NPY/AgRP expressing neurons in ARC and linked with a low level of leptin. In VMN, NPY binds to NPYR1 and inhibits the anorexigenic response of VMN, and further BDNF infusion to VMN, PVN, and DMN in rats inhibits food intake and body weight [226,236,237,238,239,240]. In addition, intraperitoneal administration of NPY in rats resulted in decreased expression of BDNF in the hypothalamus [241]. It is interesting to note that it is not only VMN, other hypothalamic region PVN is also involved in BDNF associated changes in CRH and regulation of downstream signaling pathways including protein kinase A, protein kinase C, extracellular signal-regulated kinases, mitogen-activated protein kinases and modulation of anorexigenic and orexigenic stimulus in the hypothalamus. Further, BDNF has also been associated with eating disorders, glucose homeostasis, and some molecular determinants linked to food intake and obesity.

In addition to the regulation of anorexigenic and orexigenic response in the hypothalamus, BDNF plays an essential role in neuronal development, neuronal survival, differentiation, synaptic neurotransmission, and synaptic plasticity as well as many neuropsychological disorders. These divergent effects of BDNF are associated with modulation of key neurotransmitters in the brain and indicate the role of BDNF in the regulation of food intake by interrupting neurotransmission that not is only limited to hypothalamus but is also found in some other brain regions including the cortex, hippocampus, and reward area. In this context, Pelleymounter et al. described that infusion of BDNF in rats resulted in an increased ratio of 5 hydroxyindole acetic acid and serotonin (5HIAA/5HT) in the hypothalamus that leads to suppressed appetite and body weight [242]. Furthermore, mice deficient in BDNF exhibit perturbed 5HT not only in the hypothalamus but also in the hippocampus and cortex, and it plays a determinant role in serotonin-mediated regulation of food intake and satiety [243]. The functional significance of BDNF in regulation of DA neurons and its impact on hedonic feeding via modulation of mesolimbic system has been demonstrated [244]. The region-specific ablation of BDNF in VTA resulted in the consumption of palatable high-fat food when compared to control mice with no changes in the consumption of normal standard chow. Mice lacking BDNF exhibit loss of evoked DA release from VTA terminal in NAc supporting the loss of DAergic neuronal activity and neurotransmitter release. It was argued that overeating is a possible compensatory mechanism to restore the suppressed DA. In parallel to BDNF deficient mice, leptin-deficient mice also exhibit suppressed evoked DA release in NAc [245]. DA receptor1, agonist blocks over eating behaviour in BDNF deficient mice [234].

The role of BDNF in food intake, appetite, energy homeostasis, and obesity is undisputed; however, the mechanism and mode of action of BDNF in such a complex process is not well understood. Most importantly, how SST might be involved in the regulation of BDNF mediated function in obesity is not known yet. In this perspective, the functional role of SST via activation of different SSTR subtypes in the modulation of BDNF and regulation of obesity is well supported by the regulation of hypothalamic–pituitary–adrenal axis (HPA) in presence of BDNF, CRH, and glucocorticoids (GCs). There are several possible mechanistic explanations by which BDNF and SST may affect obesity and energy homeostasis. Previous studies have shown decreased SST expression at molecular levels in BDNF KO mice or in mice with disrupted BDNF transcript [246,247,248,249]. A dose-dependent increase in SST expression has also been reported in presence of BDNF and neurotrophin-3 but not with nerve growth factor [250]. BDNF is known to enhance SST mRNA expression probably through SST expressing neurons in the hypothalamus as well as in cortical cells [224,250]. With this well-established role of BDNF, it can be argued that the absence of BDNF lessen SST effect in obesity independent of inhibitory action of SST in the regulation of key hormones involved in obesity. SST and leptin exert opposing effect on appetite in contrast to reciprocal roles between leptin and BDNF in hypothalamus, indicating indirect relation between SST and BDNF in regulation of eating behaviour and obesity.

## 5. Role of Somatostatin and Somatostatin Receptors in Stress and Anxiety Mediated Food Intake Behaviour

Pre- and post-episode of stress and anxiety contribute to food intake and suppressed desire to eat [251,252,253]. The intensity and duration of stress play a decisive role in developing several neuropsychological conditions including mood disorder, i.e., post-traumatic disorder, anxiety, and depression, with a significant impact on appetite [254]. The responses to different external stimuli that are linked to the induction of stress often demonstrate increased expression of SST and SSTR subtypes at the level of protein and mRNA in hypothalamic regions [255,256,257]. Stengel et al. described that the activation of SSTR subtypes suppressed the CRH associated with endocrine, synaptic, behavioural, and visceral response often linked to stress, attesting the role of SST [255,258]. This strengthens the concept about the pivotal role of the peptide in CRH mediated stress. In the CNS, including hypothalamus, the activation of CRH is a crucial determinant of stress-related changes in rodents and humans [259]. Previous studies using the acute tail suspension stress model described SST analogues selectivity and role of icv versus. iv route of administration on suppression of increased ACTH [256,260,261,262]. SST-mediated opposing effect depends on isoform and mode of administration on stress-induced changes; suppression of hypothalamic CRH via activation of SSTR2 and -5 inhibit stress-induced ACTH release [263]. Furthermore, potassium chloride-induced CRH release is blocked by SST, OCT, and cortistatin not only in the hypothalamus but also in the hippocampus in parallel to increased release of CRH from the cortex [257]. This effect was selective and specific to SST-28, which displayed a high affinity to SSTR5. Viollet et al. demonstrated that SSTR2 knockout (KO) mice showed an increase in ACTH, a major regulator of stress response, attesting the role of SSTR2 in the modulation of stress associated behaviour [264]. Furthermore, delayed stress related to gastric emptying and colonic motor function is also a well-established function of SST [122,165,255,259].

The activation of brain SSTR2 interferes in the prevention of CRH mediated anorexic effect [257,263]. As described above, icv administration of pan SST agonist OTD8-SST prevents an acute stress-induced decrease in food intake [188,263]. SSTR2 antagonist (OTD8-SST) abolished SST-mediated orexigenic effect in the rat following icv treatment [163]. Further in support that SSTR2 is the prominent receptor subtype in the regulation of orexigenic action, SSTR2 selective antagonist administered prior to the dark phase reduced food intake [26,164]. Such a pronounced feeding behaviour is also supported with strong SSTR2 expression in the hypothalamic brain region associated with the regulation of food intake. It is interesting to note that the orexigenic effect of SSTR2 not only prompted food intake but also resulted in an increased number of the meals by reducing the intervals and meal size, suggesting the role of SSTR2 in enhanced food consumption by blocking satiety without any effect of satiation [209].

The expression of SSTR2 in the brain regions involved in anxiolytic effect in the stress model of anxiety support its role in anxiety. Furthermore, evidence also exists from behavioural studies showing that SSTR2 specific agonist but no other subtypes suppressed anxiety [256,263]. SSTR2 mediated effect has also been associated with membrane hyperpolarization and decrease in input resistance and suppressed neuronal excitability in the brain [161]. Taken into consideration, it is well established that CRH exerts a crucial role in the anxiolytic response to stress, and SSTR2 KO mice in response to stress exposure displayed augmented release of CRH-ACTH [255,256,258]

## 6. Somatostatin, Glucocorticoids and Food-Seeking Behaviour

GCs are prominent hormones and a crucial player in the regulation of metabolic homeostasis. In addition to the well-established role of GCs in the regulation of GH synthesis and release as well as in inflammatory and cardiac function, they are also linked to several behavioural and pathological conditions including anxiety, depression, and post-traumatic stress disorder [265,266,267]. Most of these physiological and behavioural actions of GCs are mediated by a nuclear receptor, namely glucocorticoid receptor. Several recent studies have implicated GCs in obesity and insulin resistance. Therefore, the main focus of this section is the role of GCs in the regulation of food intake and obesity, and we further discuss the role of SST in association with GCs in appetite. GCs’ effect is not limited to the peripheral system, but they are actively involved in the regulation of synthesis and release of several neuropeptides from the hypothalamus and elicit a profound effect on food intake and energy expenditure [268,269]. The stimulation of NPY and inhibition of CRH as well as melanocortin release is believed to be the plausible mechanism in the promotion of food intake with increased GCs [269].

Similar to CRH, stress-associated changes in GCs affect ACTH levels that participate in appetite via stimulation of NPY/AgRP [254]. Most appetite associated responses of GCs involves insulin in a distinct manner depending primarily on intensity. In acute conditions, GCs act on the pancreas and stimulate secretion of insulin that exerts an inhibitory role on food intake, whereas chronic exposure to GCs leads to insulin resistance, i.e. the loss of insulin’s ability to inhibit NPY/AgRP in ARC. In a condition such as Cushing’s syndrome, increased appetite, weight gain, and insulin resistance are associated with excess GC. An additional mechanism for the role of GCs in the regulation of food is the release of ghrelin from the gut that acts on pituitary and brain and modulates the negative feedback of ACTH and GCs. Furthermore, under chronic and severe stress, increased GCs secretion resulted in increased circulatory ghrelin which activates NPY/AgRP and consequently food intake. It is interesting to note that increased insulin with GC also works on DAergic neurons in VTA and suppresses the rewarding nature of food. In a classical feedback loop, GCs enhanced leptin, which by negative mechanism inhibits GC. Acutely increased leptin in the response of GC exert anorexic effect via activating CRH and by inhibiting NPY release, whereas chronic leptin prevents activation of HPA-increased GH by inhibiting NPY release. The exact role of SST in the regulation of GCs release and its effect on food intake is not well understood. However, it should be noted that SST release is stimulated by GCs in a dose-dependent manner, displaying biphasic effect with stimulatory effect in presence of low doses and opposite at higher doses [3]. Such differential expression level of SST might impact appetite accordingly. In addition to the regulation of stress associated food-seeking behaviour by SST and GCs, indirect association between hypothalamic peptide and neurotransmitters, peripheral ghrelin release, and interaction with DAergic neurons are compelling pieces of evidence to propose the role of SST in regulation of GCs-associated increased weight gain, insulin sensitivity, and adiposity.

## 7. Role of Somatostatin in Regulation of Insulin, Leptin, POMC, AgRP and Ghrelin Induced Signaling on Food Intake

SSTR subtypes are well expressed in different nuclei of the hypothalamus and colocalized with SST and tyrosine hydroxylase as well as cannabinoid receptor 1 in receptor and region-specific manner [104,166,270]. The most prominent players identified anatomically and physiologically in peripheral as well as in CNS which play a positive and/or negative role in the regulation of food-seeking behaviour include leptin, POMC, AgRP, ghrelin, and SST [271,272]. However, the molecular details and signaling pathways associated with the determinant role of these hormonal peptides in a complex process for orexigenic and anorexigenic stimuli is still not well understood. The different composition of hormone-producing and responding cells in hypothalamic nuclei relay orexigenic and anorexigenic stimuli by stimulating the release of leptin, POMC, AgRP, ghrelin, and SST [273,274,275].

Previous studies have shown the role of insulin and leptin (two prominent members of adiposity negative-feedback model of energy homeostasis) towards stabilization of fat stores by well-organized information to the brain through circulating signals [55,64]. The role of insulin in food intake was proposed years before the concept of leptin [276]. Insulin is the adiposity signal involved in food regulation; it senses body fat and is released in proportion to that. Previous studies have demonstrated that insulin is the first hormonal signal which is associated in the regulation of body weight through CNS [277,278]. Insulin from circulation reaches to the brain and results in suppression of energy [279,280]. Most importantly, insulin receptor-positive neurons have been shown in most brain regions linked to food intake. Insulin administration to the brain is associated with lesser food intake in contrast to observations in lack of the hormone. Obici et al. confirmed the importance and functional significance of hypothalamic insulin by using antisense oligonucleotides for insulin receptors, which, when injected to the third ventricle, leads to hyperphagia and increased expression of NPY/AgRP [281]. Further studies from Choudhary et al. showed that, in insulin substrate 2 deficient mice, POMC neurons were not affected [282]. Although insulin displays appetite inhibition following application to the brain, the physiological significance of such insulin effect and molecular mechanism is still elusive. The roles of SST in inhibition of insulin release from the pancreas and several other hormones from GIT which are involved in a complex process of food intake are worth mentioning here. Consistent with previous studies, insulin stimulates SST release from the hypothalamus in contrast to inhibition from pancreas and gut [3]. Glucose is known to inhibit SST from hypothalamus, whereas most aminogenic stimuli are devoid of any effect. Moreover, in the gut, SST secretion is prompted by luminal without any effect of nutrients in circulation [3]. Body fat store is critical in reduced insulin sensitivity and previous studies have shown that, to compensate insulin resistance in normal glucose homeostasis, insulin secretion must increase in basal and post-meal as weight increases [64,283]. With these observations, it was proposed that in such cases impaired pancreatic β-cells response might cause hyperglycemia, which further leads to type 2 diabetes and obesity. The authors also emphasized that gradual progression of obesity with increased insulin secretion supports enhanced insulin to CNS and consequent regulation of weight gain [64].

Leptin is primarily confined and released from adipocyte and exerts a crucial role in the regulation of body weight, appetite, and energy homeostasis, especially in anorexigenic behaviour [284]. Furthermore, the role of leptin is supported by hyperphagia and obesity in db/db mice with mutation in leptin. Previous studies have shown the loss of leptin action in obese cases, indicating leptin resistance and many conditions have been reported exhibiting mutation in leptin receptor leading to obesity due to some other reason in mice and rats [64]. Hypothalamus is composed of complex and heterogeneous neuropeptide pathways with anabolic and catabolic effector response and is involved in regulation of orexigenic and anorexigenic stimuli. Amongst them, NPY and melanocortin are prominent peptides that, via interaction with leptin, play food stimulatory and suppression role. Whether SST, which plays an essential role in obesity, is linked to leptin is not well understood. The opposing effect of SST (orexigenic) and leptin (anorexigenic) raised question of whether these two peptides interact with each other and antagonize feeding behaviour. SST infusion displayed inhibitory effect on circulation levels of adipokines including leptin in lean subjects with no effect in obese patients [285]. The most likely association between SST and leptin emerged from studies by Simon et al. describing SST binding with lipolytic action in adipocyte and distribution of SSTR1, -3, and -4 mRNA in human adipocytes [173,286]. Although the direct evidence supporting such interaction is largely awaited, existing indirect observation supports such notion, through modulation of leptin induced signaling pathways such as signal transducer and activator of transcription proteins (STAT), via activation of receptor protein (NPY, OX, opioid, and cannabinoid receptors), or by using DR as reward system associated with food-seeking behaviour [26]. In the hypothalamus, leptin inhibits SST secretion, whereas SST regulates leptin action negatively in the hypothalamus [87,287,288]. Following icv administration of leptin, increased density of SSTRs has been shown on Day 6 but not on Day 1 due to enhanced expression of SSTR2 at the levels of mRNA and protein. Such a prominent effect of SSTR2 was also accompanied by increased activation of insulin signaling and changes in c-JUN and cAMP response element binding, which were blocked by leptin antagonists [289]. In parallel to these observations, the administration of SST or SSTR specific agonists in combination with leptin diminished the activation of STAT and its nuclear translocation in hypothalamus including VMN, DMN, ARC, and LHA. Amongst SSTR agonists, SSTR3 displayed prominent effect followed by SSTR1 and SSTR2 [287]. Studies have also demonstrated reduced serum leptin levels in humans in response to peripheral treatment with SST and its analogues [290,291,292]. The reduced SST mRNA and basal levels in response to leptin administration to hypothalamic fetal rat neurons implicates that leptin regulates hypothalamic *SST* gene expression [288,293]. Stengel et al. described the role of SSTR2 in SST mediated orexigenic effect [26,142,164]. Leptin and SST interaction involving SSTR2 is further supported by observation demonstrating increased SSTR2 expression at the level of mRNA and protein following acute central infusion of leptin [287,289]. The authors also proposed that changes in SSTR effectors system may antagonize the leptin effect. Stepanyan et al. demonstrated that SST-14 affects central leptin-signaling via two mechanisms: reduced STAT3-activation and leptin-mediated anorexia [287]. In vivo administration of SST-14 through icv route antagonizes the 24-h anorexigenic effect, which was reported to be mediated by SSTR1, SSTR2, and SSTR3 [287]. The presence of leptin receptors has been reported with relatively high expression in inhibitory GABAergic in comparison to excitatory glutamatergic neurons in hypothalamic regions including VMN, ARC, DMN, and LHA [86,294,295,296,297]. Consistent with the well-established notion that SST inhibits the secretion of the pancreatic hormone as well as all hormones from GIT which usually inhibit adiponectin, a direct role of SST in this direction is highly predictable. These results suggest that leptin and SST play opposing roles on food intake, obesity, and energy expenditure (Figure 4).

POMC and AgRP exert negative and positive effects on food intake, respectively [69,298,299,300]. Lohr et al. recently reported functional cross-talk between POMC and SST and proposed that POMC neurons at the level of HPA are involved in the stimulation of hypothalamic SST neurons [301]. Furthermore, previous studies in fasted hungry mice have revealed the opposing role of leptin in the regulation of POMC (inhibition) and AgRP (activation) expressing neurons in ARC [302,303,304,305,306]. These observations indicate an interaction between SST and POMC with a possible impact on food-seeking behaviour.

The activation of AgRP positive neurons in ARC prompt response in food-seeking behaviour is contrary to the lack of food desire in absence or inhibition of AgRP neurons [89,91,92,212,298]. Whether AgRP mediated orexigenic behaviours involves SST is not well understood. Campbell et al. recently demonstrated the presence of SST and AgRP in the same neurons in ARC [213]. The activation of neurons displaying colocalization enhanced feeding incentive that is the indication of distinct SST positive neurons associated with different food-seeking behaviour with no selective preference for fat [307]. These results suggest a direct relation between SST and AgRP in food-seeking behaviour. Furthermore, neuronal cell population in ARC expressing NPY/AgRP is under the inhibitory influence of leptin, whereas insulin is associated with food intake upon activation. In contrast, neuronal cells expressing POMC suppressed food intake and are stimulated by leptin and insulin (Figure 4) [308]. Taken together, these results indicate a complex interaction between SST and orexigenic and anorexigenic neuronal circulatory system in the hypothalamus. Although controversies exist, the above is compelling evidence in support of possible interaction between these crucial determinants of orexigenic and anorexigenic stimulus at the cellular levels.

Ghrelin, an orexigenic hormone, is prominently expressed in gastric mucosa [309]. The first possible indication for the interaction between ghrelin and SST emerged from studies showing that ghrelin mediated release of GH is blocked by iv administration of SST in healthy volunteer as well as in pituitary cells not only by inhibiting SST release but also directly by activating GH release [310,311]. SST and ghrelin exert an antagonizing effect in the regulation of GHRH release from ARC. Chronic infusion of ghrelin plays a critical role in the regulation of body weight, which is further supported by the loss of body weight and increased energy expenditure in ghrelin and ghrelin receptor double KO mice. As described by Korbonits et al., ghrelin works via three different mechanisms to regulate the appetite [312]. First, ghrelin released from the stomach may cross BBB and act on the ghrelin receptor present in the hypothalamus. Second, ghrelin is produced in the hypothalamus locally and implicates a direct effect on hypothalamic nuclei. Third, ghrelin is carried to the brain via vagal nerve and NTS to produce an orexigenic effect [312]. Stengel et al. reported opposing effects of SST in central and peripheral tissue on basal plasma levels of ghrelin via activation of SSTR2 [142]. The role of cortistatin, which binds to SSTR subtypes, has also been proposed on ghrelin release [142]. Earlier studies have shown the direct role of cortistatin in ghrelin release through interaction with ghrelin receptors, indicating a possible association between ghrelin and SST [142,313]. Further in support, studies have also described that gastric ghrelin synthesis and stimulation are physiologically inhibited by endogenous SST [314,315]. Since ARC contains NPY/AgRP and POMC, leptin, and insulin which inhibit NPY/AgRP eventually inhibit food intake, they thus affect energy expenditure and body weight. On the other hand, leptin also stimulates POMC and hence reduces food intake (Figure 4) [52,316]. Ghrelin activates NPY, AgRP and OX expressing neurons and inhibits POMC and CRH producing neurons [304,317]. A recent case report from a patient suffering from leptin deficiency demonstrates the crosstalk between gut hormones [318]. This study highlighted the role of leptin substitution in leptin deficient patients, which facilitates significant elevation in meal-stimulated insulin, peptide YY, and GLP-1 levels. Furthermore, the levels of ghrelin were reduced with concomitant weight loss, indicating the role of leptin in regulating ghrelin secretion and improving central satiety [318]. Taken together, these observations specify different and opposite effects of ghrelin and leptin on hypothalamic neurons in regulation of food-seeking behaviour.

## 8. Role of Somatostatin, nNOS and Cannabinoid Receptor in the Regulation of Appetite

Several overlapping physiological functions including appetite and comparable distributional pattern of SST, nNOS, and CB1R in the hypothalamus are compelling evidence to debate the possible interaction between these prominent orexigenic players in rodents and human. In agreement with previous studies and our recent observations with colocalization of CB1R and SST in different hypothalamic regions involved in food intake, we speculate a collaborative function between SST and CB1R in food intake behaviour [319,320,321]. The stimulation of food intake following endocannabinoid injection in VMN that is blocked in the presence of CB1R antagonists and increased level of endocannabinoids is seen in the hypothalamus during fasting, which is maintained at a normal level following food consumption [322]. The role of endocannabinoid independent of CB1R has also been reported, but whether SST contributes in such role of endocannabinoid in food intake is not known. However, previous studies have shown weight loss and inhibition of food intake in rodents upon blocking CB1R and also the weight-reducing role of CB1R antagonist [71,323]. Similar to SST and CB1R, the role of nNOS is well established in feeding, which is further supported by decreased food intake and weight loss upon inhibition of nNOS and increased activity of nNOS in lack of food [324]. Most importantly, appetite associated hormones that are inhibited by SST are also modulated by nNOS. The opposing effect of ghrelin and NPY and leptin in the hypothalamus also support interaction with leptin, ghrelin, and NPY beyond SST [325,326]. The inhibitors of nNOS suppressed ghrelin induced feeding and nNOS are needed in leptin mediated regulation of energy control [325,327]. In the quest to develop more specific and safe pharmacological interventions to treat obesity, the recently presented molecular ultrastructure of human CB1 holds potential for newer opportunities for design of novel anti-obesity drugs [328,329]. Hence, allosteric agents directed against CB1 such as hemopressin or pregnenolone may be formulated newly with significantly improved side effect profile [330,331,332,333,334]. Finally, another pharmacological approach aimed at selective blockade of peripheral CB1, which proved metabolic benefits independent of modification in feeding behaviour [335,336,337]. Therefore, understanding SSTRs and CB1 dependent signaling are relevant for development of safe pharmacological interventions for the treatment of obesity.

## 9. Functional Cross-talk Between Receptor Proteins: A Mode for the Promotion of Satiety or Obesity: Future Perspective

Mounting evidence supports the role of many receptor proteins including the receptors for SST, endocannabinoid, serotonin, epinephrine, DA, GABA, glutamate, and opioid in the regulation of appetite and energy homeostasis. These receptors have shown selective and preferential co-expression as prominent players of anabolic and catabolic function including NPY/AgRP, leptin responsive neurons, POMC, MC3R, and MC4R in different nuclei of hypothalamus including ARC, PVN, VMN, and LHA. However, the distribution of these receptors is not limited to the hypothalamus but also exists in other brain regions including the hippocampus, cortex and most importantly reward pathway, i.e., VTA-NAc, the brain region associated with food intake reward behaviour. Despite the profound impact on appetite associated hormonal secretion and regulation, currently nothing is known on whether these receptors functionally interact with each other and function in concert in food control in areas such as the hypothalamus, which needs to be addressed. Furthermore, the requirement of the N-methyl-D-aspartate receptor in the activation of AgRP neurons during fasting is important but uncertain physiologically in food-seeking process specifically in fasting [338]. Mounting evidence describes the protein–protein interaction between a member of GPCR family and SSTRs, DRs, opioid receptors, adrenergic receptor, and cannabinoid receptors, and their homo- and/or heterodimerization is well studied [339,340,341,342]. These prominent members of the GPCR family exert a crucial role, directly and/or indirectly, in the regulation of food-seeking behaviour [343]. Furthermore, SST and NPY have been shown to be expressed in the same neurons in the rat brain and speculate cross talk between SSTR and NPY receptors in the brain in addition to SST mediated inhibition of GABA [344]. Nothing is currently known on whether different receptor proteins functionally interact in the hypothalamus and GIT and work in concert in the regulation of food intake. The concept of receptor dimerization and its role in food intake behaviour and perturbed energy homeostasis emerged from previous studies showing that icv administration of SST in rat induced stimulation of food intake is associated with NPY1, μ-opioid receptors and OX1 mediated orexigenic signaling pathways [94,121]. Leptin acts on DAergic neurons in VTA to suppress feeding and insensitive to palatable food and mice lacking D2R are more sensitive to leptin. These studies further strengthen the concept regarding the role of receptor interaction in obesity and satiety via regulation of different hormonal secretion and release, in response to either central or peripheral input, as described in several pathological conditions via modulation of the signal transduction pathways [15,69,345]. Our recent observations from the colocalization of SST and CB1R in the hypothalamus further suggest possible interaction between SSTR subtypes and CB1R [104]. We also described an interaction between CB1R and SSTR5 in the brain [270]. Inhibition of GABA release and glutamatergic neurons by SST and CB1R indicate the differential role of SST and CB1R in the regulation of food intake via modulation of inhibitory and excitatory neurotransmission in the hypothalamus. Such speculation further supports the stimulation of appetite associated with the activation of excitatory neurons in LHA [122]. In addition to serving as a prominent neurotransmitter in CNS, serotonin exerts inhibitory effect on appetite. The role of central and peripheral serotonin is well established in appetite associated obesity and energy homoeostasis. The agonists for serotonin receptor are in use for the treatment of obesity. In this context, association between serotonin receptors in anorexigenic and orexigenic neurons in hypothalamus specifically in neuronal cells expressing GABA receptors and SSTR subtypes is highly expected. Taking this into consideration, expected cross talk amongst key receptor proteins as proposed here might help in synthesis of effective therapeutic avenue in regulation of obesity and associated devastating pathologies including diabetes and cardiac complication.

## 10. Somatostatin, Satiety, and Obesity—What Is the Link?

The expression of SST in hypothalamic regions including ARC, VMN, and LHA and D cells of GIT demonstrating its role in food-seeking behaviour and in the regulation of hormonal secretion as well as gut motility are compelling pieces of evidence to support the role of SST in satiety and obesity (Figure 4). There is mounting behavioural, morphological, biochemical, and pharmacological evidence supporting the role of SST in food intake associated with obesity. Hyperinsulinemia due to hypothalamic dysfunction may increase adipogenesis and lead to weight gain [346]. In this context, current evidence suggests that SST and its analogues, through binding to different SSTR subtypes on the pancreatic β-cells, decrease the release of insulin, which may be beneficial in the treatment of obesity [209,215,216,218,219,220]. SST treatment can suppress weight gain in children with hypothalamic obesity but has minimal effect on weight loss in adults with obesity [20,201,209,217]. In patients with obesity, SST induces satiety through multiple mechanisms and is beneficial for weight management [180]. SST and its analogues exert their effects directly in obesity through the insulin-mediated mechanism and indirectly by inducing satiety.

## 11. Conclusions

In the regulation of satiety, various contradictory mechanisms are involved and are still not fully understood. However, the involvement of various receptors, satiety factors, and mechanisms of various pathological conditions linked to the regulation of satiety are tempting to explore further in details. Recently, the development and synthesis of new receptor agonists and antagonists provide a broader spectrum for the role of SST in several pathological conditions including appetite and satiety. SST induces satiety by delaying GI motility and emptying. SST mediated inhibitory action on satiety associated factors such as CCK may decrease the effect on satiety. On the other hand, the central action of SST on SSTR2 in the brain stimulates food and water intake. SST analogues, with their broad array of actions, are better candidates for the treatment of obesity and future studies might uncover the new role and mechanism in certain diseases, which lead to the loss of appetite. There is a scope for the development of receptor-specific agonists and antagonists for selective therapeutic intervention in food-seeking behaviour. In several pathological conditions, including tumours of different origin, HIV, etc., the loss of appetite is the major problem while SST and its analogues provide a potential therapeutic option. In addition to hypothalamus, elucidating the role of other efferent and afferent projections of the brain stem, cortex, and reward system as well as the exploration of multiple downstream signaling pathways involving SSTR subtypes in regulation of anabolic and catabolic hormone will shed new light on food-seeking behaviour.

## Figures and Tables

**Figure 1 ijms-21-02568-f001:**
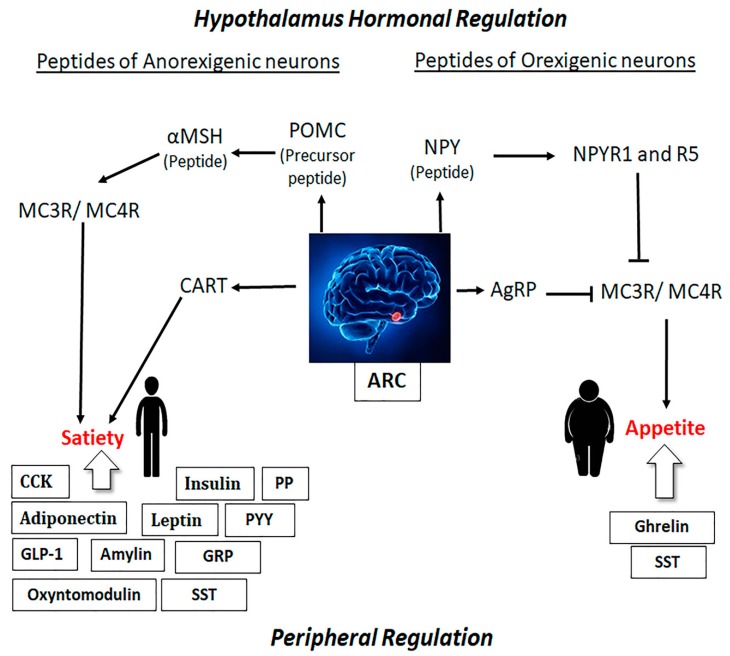
Overview of orexigenic and anorexigenic stimuli in association with peripheral hormones in regulation of appetite and satiety. ARC is composed of orexigenic neurons producing NPY and AgRP and anorexigenic neurons, which synthesize POMC and CART. The peripheral hormones regulate appetite via endogenous gut hormones such as CCK, GLP-1, oxyntomodulin, PYY, and PP, producing an anorexigenic effect. Ghrelin and SST have an orexigenic effect in humans and prompt food intake. In addition, SST produces dual effect (orexigenic and anorexigenic). Pointed arrows indicate activation; Blocked arrows indicate inhibition. ARC, arcuate nucleus; AgRP, agouti related peptide; CART, cocaine and amphetamine-regulated transcript; CCK, cholecystokinin; GLP-1, glucagon like peptide-1; GRP, gastrin-releasing peptide; NPY, neuropeptide Y; POMC, propiomelanocortin; PYY, peptide YY; NPY, neuropeptide Y; PP, pancreatic polypeptide; αMSH, α-melanocyte stimulating hormone; MCR, melanin-concentrating hormone receptor; SST, somatostatin.

**Figure 2 ijms-21-02568-f002:**
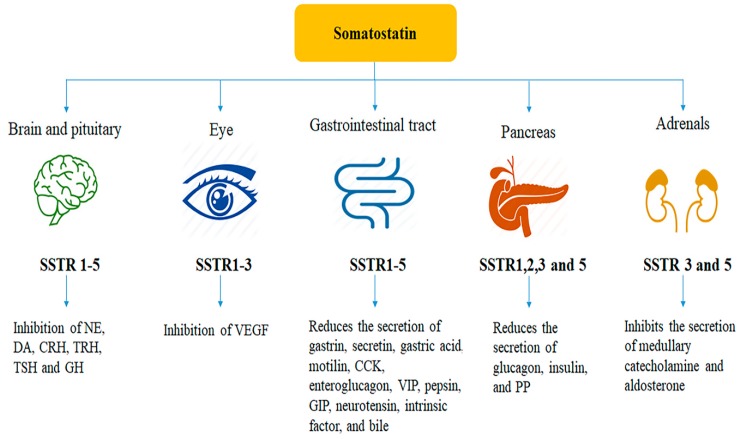
Schematic presentation showing somatostatin receptors associated with different roles of somatostatin in peripheral tissues. CCK, cholecystokinin; CRH, corticotropin releasing hormone; DA, dopamine; GH, growth hormone; GIP, gastric inhibitory polypeptide; NE, norepinephrine; PP, pancreatic polypeptide; TRH, thyrotropin releasing hormone; TSH, thyroid-stimulating hormone; VEGF, vascular endothelial growth factor; VIP, vasoactive intestinal peptide.

**Figure 3 ijms-21-02568-f003:**
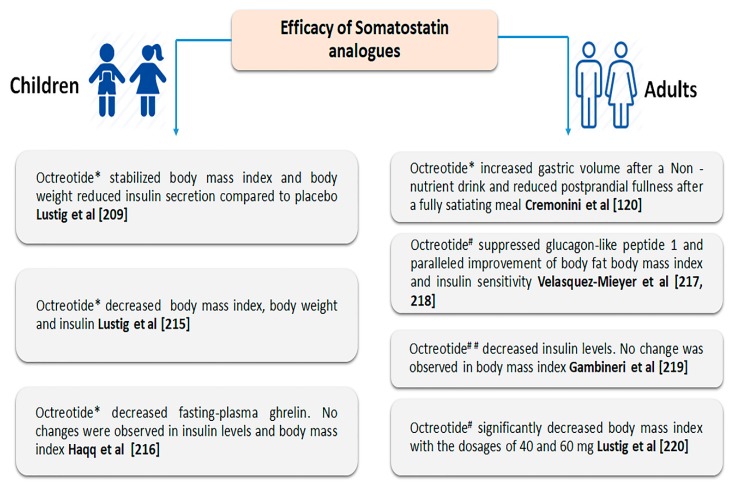
Comparative assessment of somatostatin analogues in the treatment of obesity in adults and children with different route of administration. * Subcutaneous; ^#^ long-acting release (intramuscular); ^##^ long-acting drug with low-calorie diet.

**Figure 4 ijms-21-02568-f004:**
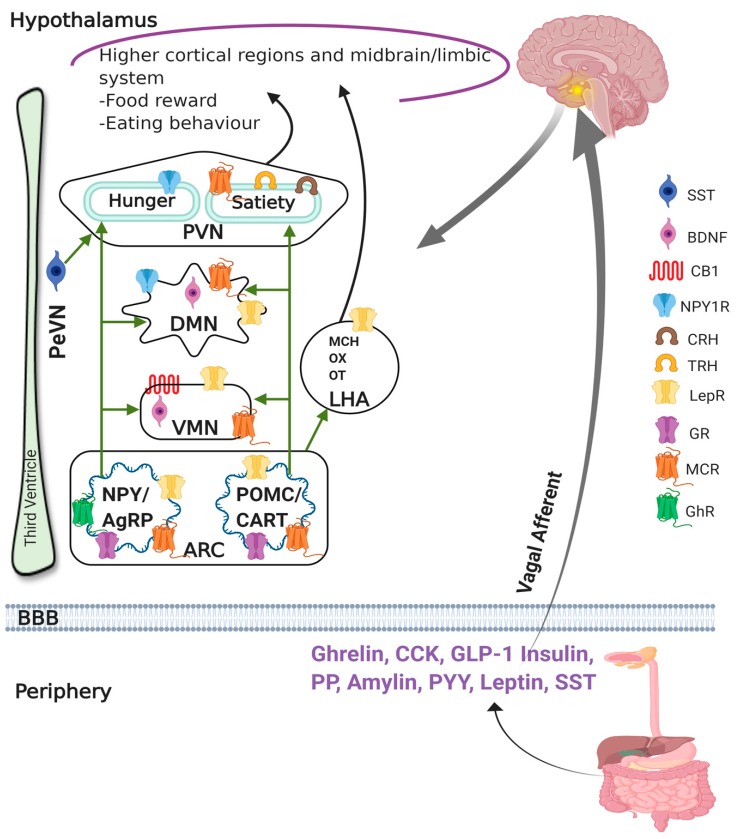
Schematic illustration of hypothalamic regions and peripheral hormones in regulation appetite and satiety. NPY/AgRP and POMC/CART neurons in the ARC project to second-order neurons in other hypothalamic nuclei, including the PVN, DMN, VMN, and LHA. These second-order hypothalamic neurons express anorexigenic neuropeptides (CRH, TRH, OT, and BDNF) and orexigenic neuropeptides (OX and MCH), which regulate appetite and modulate energy homeostasis. In addition, the regulation of energy balance involves an integration of signaling from the hypothalamus, brain stem, and reward pathways of the mesolimbic system. SST acts via hypothalamic nuclei and pancreas. Leptin acts directly on the NTS as well as hypothalamic nuclei, and it can modulate appetite through different pathways. Green arrows indicate activations within hypothalamus; Black arrows indicate activations other than hypothalamus. ARC, arcuate nucleus; AgRP, agouti related peptide; BDNF, brain-derived neurotrophic factor; CART, cocaine- and amphetamine-regulated transcript; CB1, cannabinoid receptor 1; CCK, cholecystokinin; CRH, corticotropin, releasing hormone; DMN, dorsomedial nucleus; GhR, ghrelin receptor; GLP-1, glucagon like peptide-1; GR, glucocorticoid receptor; POMC, propiomelanocortin; PYY, peptide YY; IRN, insulin receptor neuron; LHA, lateral hypothalamic area; LepR, leptin receptor; MCH, melanin-concentrating hormone; MCR, melanin-concentrating hormone receptor; NTS, nucleus of the tractus solitaries; NPY, neuropeptide Y; OT, oxytocin; PeVN, periventicular nucleus; PVN, paraventicular nucleus; SST, somatostatin; TRH, thyrotropin-releasing hormone; VMN, ventromedial nucleus. Created with paid subscription of BioRender.com.

**Table 1 ijms-21-02568-t001:** The summary of the effects of gastrointestinal hormones on food intake.

Peptide Hormones	Origin of Secretion	Binding Receptors	Effect on Food Intake	References
Cholecystokinin	I cells	Cholecystokinin receptor subtype A	Decreased	[114]
Glucagon-like peptide 1	L cells	Glucagon-like peptide-1 receptor		[115]
Pancreatic polypeptide	PP cells	Pancreatic polypeptide Y2 & Y4 receptor		[116]
Peptide tyrosine tyrosine	L cells	Peptide tyrosine tyrosine Y2 receptor		[117]
Oxyntomodulin	Oxyntic cells	Oxyntomodulin		[118]
Somatostatin	Delta cells; Neurons	Somatostatin receptors	Increased/Decreased	[26,119,120,121,122]
Ghrelin	P/D1 and Epsilon cells	Growth hormone secretagogue receptor	Increased	[123]
Leptin	Fat cells	Ob-Rb, Leptin receptor	Decreased	[124]
Adiponectin	Adipose tissue	Adiponectin receptor 1 and 2		[125]
Insulin	Beta cells	Insulin receptors		[126]
Amylin	Beta cells	Amylin-specific receptors		[127]

**Table 2 ijms-21-02568-t002:** Summary of studies for the effect of SST analogues and receptor subtype on food intake.

Treatment	Species	Route of Administration	Proposed Mechanism	Effect on Food Intake	References
SST	Rat and baboon	IP	Vagally mediated mechanism	Decreased	[174,183]
OCT	Rat	ICV	Activation of brain SSTR2	Increased	[184,185]
SST	Healthy volunteers	IV infusion	Reduced gastric emptying and gut motility	Initially decreased Increased in presence of low dose of intraduodenal fat Post meal satiety was higher	[25]
SST-14	Chick	ICV	Activation of brain SSTR2	Increased	[186]
SST analogue (ODT8-SST)	Rat	ICV	Activation of brain SSTR2	Increased	[163]
OCT	Mouse	ICV	Activation of brain SSTR2	Increased	[164]

ICV, intracerebroventricular injection; RCT, randomized controlled trial; IP, intraperitoneal; IV, intravenous.

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
