# Peer review of "Role of Somatostatin in the Regulation of Central and Peripheral Factors of Satiety and Obesity"

_ijms, 2020, doi:10.3390/ijms21072568_

Round 1

Reviewer 1 Report

In the paper “Role of Somatostatin in the Regulation of Central and Peripheral Factors of Satiety and Obesity” the authors’ intent is to provide a comprehensive review on the role of SST in modulating the central and peripheral signals of control of satiety and obesity.

The topic is interesting and relevant, given that obesity represents a risk factor for cardiovascular and metabolic diseases. However, the review is not well organized and sometime confused. Furthermore, a very similar review has been recently published on “Endocrine review” https://doi.org/10.1210/er.2018-00283 (not cited)

Please, find enclosed some tips.

The review is quite long, and not always logical. A review more focused on SST and its receptors would likely catch the authors’ attention and better convey the novelty.

The first part (paragraph 1 and 2- pagg 3-15) is very long and not particularly innovative. A summary of the central and peripheral factors involved in the control of appetite and satiety would be preferable. Figures and schemes can be helpful.

The paragraphs on BDNF and GC are not clear in the context of the review (sst effects on..)

In some points, the message is not clear, and often the same concept is repeated twice. Ex Pages 5, rows 120-124 .

A revision of the English style is required. There are several non-sense sentences. Ex. Row 485-487. Although, due to the limited understanding on the subject future studies are warranted, certain pathological condition often linked to appetite and food intake such as type II diabetes, obesity, and metabolic abnormalities described in periodontitis ?

Figure1: in the right part of the figures the authors report the pathways involved in the control of Obesity. I would substitute “Obesity” with “food seeking” or “appetite”. Indeed, the described pathway is active also in physiological conditions (exactly as satiety). Only the unbalance between the two opposite behaviours causes obesity.

Figure 2: Description of SST function in brain and pituitary (green box), there is a repetition

Tab 2 can be simplified: the first two rows (SST) are identical, the two references can be reported together in the same box (references)

Figure 6 is rather complex and can be simplified

The review recently published on “Endocrine review” https://doi.org/10.1210/er.2018-00283 has not be considered in the reference section

Author Response

We thank reviewer for his/her support and critical comments as well as for time for thorough reading of our review article.

Response to reviewer 1

  1. Reviewer noticed that review is quite long and not always logical and should be focused on SST and its receptors.

Response: We agree with reviewer. However, the role of Somatostatin (SST) and Somatostatin receptor (SSTR) subtypes has been discussed earlier but has not been addressed with reference to peripheral appetite signal. Accordingly in this review, we have discussed the role of SST in regulation of central and peripheral stimuli for appetite.

  1. Reviewer mentioned that in first part, paragraphs 1 and 2 are very long

Response: In this revised version, we have shorten these paragraphs accordingly.

  1. The paragraphs on BDNF and GC are not clear in the context of the review.

Response: Glucorticoids and BDNF exert critical role in appetite and are also crucial in regulation of SST but such interaction has not been discussed in regulation of food intake previously and this tempted us to incorporate such information in this review.

  1. A revision of the English style is required and also noticed that row 485-487 does not make much sense.

Response: This revised version has been edited and proof read extensively by colleagues. Rows 485-487 have been deleted in this revised version.

Figure 2, Table 2 and Figure 6 have been modified and simplified as suggested.

As suggested by the reviewer, “Endocrine review” https://doi.org/10.1210/er.2018-00283 has been cited in this revised version of our review along with some other references.

Reviewer 2 Report

This paper is a review on hormone regulation of food intake and somatostatin with a focus on the hypothalamus. The text is well written and clear. Below are minor edits.

Figure 1 is a bit basic on the pathway and the symbols are difficult to read as some are slanted. Instead the arrows should have a blunted end for hormones that decrease something in a pathway and the pointed arrow should be used for increasing within the pathway. What about feedback within these pathways? Also under satiety and obesity there is an arrow pointed up, which could either mean the hormones under the arrow increase satiety/obesity or satiety/obesity is increased due to the pathway. Please make this clear.

Authors throughout should be replaced with scientific investigators or another term that is more scientific. Or the authors name themselves.

For all titles use sentence case or be consistent. Table 1 Effect should be lowercase. Both table1 and 2 are very light on references.

Figure 4 is a good start, but could use some improvements. It has the same problem with '+' and '-' symbols on a slant. See my comments for figure 1. This figure is very busy and although I like the message, it is very difficult to read. There is too much color that is not helpful to explain the diagram and I don't understand the flow in reading it. Perhaps all the '-' and '+' structures could be combined respectively. I don't understand why some are faded and there is an outline around several structures. Delete what is not necessary here to tell your story.

Author Response

We thank reviewer for his/her support and critical comments as well as for time for thorough reading of our review article.

Response to reviewer 2

We thank reviewer for comments that “The text is well written and clear”

  1. ‘Authors’ throughout should be replaced with scientific investigators. For all titles use sentence case or be consistent. Table 1 Effect should be lowercase. Both table1 and 2 are very light on references.

Response: According to reviewer suggestion ‘Authors’ words have been changed throughout the review. Titles and subtitles are according to previously published review in this Journal. Table 1 and 2 has been modified.

  1. Figure 4, is a good start, but could use some improvements

Response: This figure has been improved significantly and new version of this figure is much clear and simple. Most of the colors are removed from figures.

Round 2

Reviewer 1 Report

Some modification have been introduced in the new version and the style has been improved